# Emerging Roles of Phospholipase C Beta Isozymes as Potential Biomarkers in Cardiac Disorders

**DOI:** 10.3390/ijms241713096

**Published:** 2023-08-23

**Authors:** Antonietta Fazio, Camilla Evangelisti, Alessandra Cappellini, Sara Mongiorgi, Foteini-Dionysia Koufi, Irene Neri, Maria Vittoria Marvi, Michele Russo, Alessandra Ghigo, Lucia Manzoli, Roberta Fiume, Stefano Ratti

**Affiliations:** 1Department of Biomedical and Neuromotor Sciences, University of Bologna, Via Irnerio 48, 40126 Bologna, Italy; antonietta.fazio2@unibo.it (A.F.); camilla.evangelisti@unibo.it (C.E.); alessandra.cappellini@unibo.it (A.C.); s.mongiorgi@unibo.it (S.M.); foteinidionysi.koufi@unibo.it (F.-D.K.); irene.neri3@unibo.it (I.N.); mariavittoria.marvi2@unibo.it (M.V.M.); lucia.manzoli@unibo.it (L.M.); 2Department of Molecular Biotechnology and Health Sciences, Molecular Biotechnology Center “Guido Tarone”, University of Torino, 10126 Torino, Italy; michelerusso9111@gmail.com (M.R.); alessandra.ghigo@unito.it (A.G.)

**Keywords:** phospholipase C beta isozymes, cardiovascular system, cardiac hypertrophy, diabetic cardiomyopathy, myocardial ischemia/reperfusion injury, cardiac biomarkers

## Abstract

Phospholipase C (PLC) enzymes represent crucial participants in the plasma membrane of mammalian cells, including the cardiac sarcolemmal (SL) membrane of cardiomyocytes. They are responsible for the hydrolysis of phosphatidylinositol 4,5-bisphosphate (PtdIns(4,5)P_2_) into 1,2-diacylglycerol (DAG) and inositol (1,4,5) trisphosphate (Ins(1,4,5)P_3_), both essential lipid mediators. These second messengers regulate the intracellular calcium (Ca^2+^) concentration, which activates signal transduction cascades involved in the regulation of cardiomyocyte activity. Of note, emerging evidence suggests that changes in cardiomyocytes’ phospholipid profiles are associated with an increased occurrence of cardiovascular diseases, but the underlying mechanisms are still poorly understood. This review aims to provide a comprehensive overview of the significant impact of PLC on the cardiovascular system, encompassing both physiological and pathological conditions. Specifically, it focuses on the relevance of PLCβ isoforms as potential cardiac biomarkers, due to their implications for pathological disorders, such as cardiac hypertrophy, diabetic cardiomyopathy, and myocardial ischemia/reperfusion injury. Gaining a deeper understanding of the mechanisms underlying PLCβ activation and regulation is crucial for unraveling the complex signaling networks involved in healthy and diseased myocardium. Ultimately, this knowledge holds significant promise for advancing the development of potential therapeutic strategies that can effectively target and address cardiac disorders by focusing on the PLCβ subfamily.

## 1. Introduction

Lipids and phospholipids are essential components of the plasma membrane of all mammalian cells, including the cardiac sarcolemmal (SL) membrane of cardiac muscle cells, known as cardiomyocytes. Their organization is crucial to maintain the membrane’s structure and properties, such as fluidity, stability, and selective permeability. Additionally, it regulates important cellular processes, including excitation/contraction (e-c) coupling, inflammatory responses, and cytoskeletal anchoring [1]. The influence of age and diseases can severely affect these unique properties of the SL membrane of cardiomyocytes [2]; indeed, alteration of the phospholipid profile has been associated with increased cardiovascular disease occurrence [1,3,4]. However, mechanisms underlying phospholipid remodeling in cardiac pathologies are still poorly understood.

The main phospholipids present in the SL membrane are phosphatidylcholine (PC) and phosphatidylethanolamine (PE), which represent 45% and 37% of the total phospholipid content, respectively [5]. Other detectable phospholipids in cardiomyocytes include sphingomyelin (SM), phosphatidylinositol (PtdIns), phosphatidylserine (PS), diphosphatidylglycerol, and lyso-phosphatidylcholine; each serves distinct roles in maintaining cellular structure and function.

PtdIns are a class of phospholipids that play a critical role in intracellular signaling pathways, contributing to various cellular processes [6,7] such as membrane trafficking, actin cytoskeleton organization [8], cell growth, and ion channel regulation [9]. Specific kinases catalyze the phosphorylation of PtdIns, giving rise to lipid mediators, which, in turn, undergo further modifications mediated by specific kinases, phosphatases, and phospholipase C (PLC) enzymes. As a result of these enzymatic activities, essential second messengers are generated, which play crucial roles in essential signaling cascades [9].

In particular, PLC isoforms predominantly hydrolyze phosphatidylinositol-4,5-biphosphates (PIP_2_), the most abundant phosphoinositide in the plasma membrane [10], into two second messengers: the hydrophilic acidic end-group inositol (1,4,5) trisphosphate (Ins(1,4,5)P_3_) and the neutral lipid sn-1,2-diacylglycerol (DAG). Both intermediates play a crucial role in regulating the intracellular calcium (Ca^2+^) concentration which, in turn, activates signal transduction mechanisms involved in the functional regulation of cardiomyocyte activity and the expression of specific cardiac genes [11,12]. In line with this evidence, high levels of DAG have been identified in the heart of spontaneously hypertensive rats [13], as well as Ins(1,4,5)P_3_ content in both animal and human models of heart failure (HF) [14]. Notably, a depletion of PIP_2_ has been demonstrated to be responsible for T-tubule lack and impaired Ca^2+^ handling, resulting in cardiac disorders, such as hypertrophy, HF, and diabetic cardiomyopathy [15,16,17]. Ultimately, all this evidence highlights that changes in cardiac lipid metabolism are strictly associated with the development of cardiac disorders.

Hence, this review aims to provide a comprehensive overview of the significant impact of PLC on the cardiovascular system, including both physiological and pathological conditions. After a brief introduction of the general features of myocardial PLC isozymes, the focus will be on a specific subtype of PLC, PLCβ isoforms, due to their potential contribution to various pathological processes, such as cardiac hypertrophy, diabetic cardiomyopathy (DCM), and myocardial ischemia/reperfusion (I/R) injury (Table 1). Therefore, a thorough understanding of PLCβ metabolism in cardiac tissue is crucial for gaining valuable insights into the physiology of both healthy and diseased myocardium, as well as for the development of potential therapeutic strategies.

## 2. Myocardial Phospholipases C Isozymes

PLC activation plays a crucial role in healthy and diseased myocardium by regulating the function of intracellular proteins and influencing the expression of nuclear transcription factors involved in various cellular processes [31]. The PLC family consists of six subfamilies, β, γ, δ, ε, ζ, and η, distinguished by their structures and regulatory mechanisms of activation. Each subfamily encompasses multiple isoforms and splice variants, such as PLCβ1-2-3-4, PLCγ1-2, PLCδ1-3-4, PLCε1, PLCζ1, and PLCη1-2, exhibiting distinct expression patterns across a wide cellular range [32]. Notably, recent studies have identified and classified three atypical PLCs in the human genome, leading to the classification of PLC-XD and, consequently, a total of 16 members in the PLC family [33].

In the myocardium, the principal forms expressed are PLCβ1, -γ1, -δ1, and -ε; particularly, PLCγ1 is the most abundant isozyme identified in the heart [34]. On the other hand, PLCδ1 is considered the main isozyme located at the SL membrane mainly due to the presence of basic amino acids in the N-terminal pleckstrin homology (PH) domain, which exhibits a strong affinity for PIP_2_ [35].

All PLC isoforms share a conserved core structure composed of a catalytic domain, an N-terminal PH domain, an EF hand, and a C2 domain; except for the PLCζ isoform, which lacks the PH domain within the core structure [36]. Each isoform also possesses additional regulatory domains that enable isoform-specific interactions and signaling pathways, thereby leading to diverging functions among the different PLC subtypes [36]. Additionally, PLC enzymes can be activated by different stimulations, such as tyrosine phosphorylation-dependent activation and G-protein-coupled receptors (GPCRs) through the binding of growth factors, neurotransmitters, and hormones [36]. However, it is important to note that all PLC isoforms are strictly dependent on Ca^2+^ for their activity.

In summary, cardiac PLC enzymes present a specific structural organization and localization in cellular compartments of cardiac tissue, and their activation tightly regulates various signaling pathways. Therefore, understanding the intricate mechanisms underlying their activation and regulation is essential for unraveling the complex signaling networks involved in cardiac physiology and pathology.

### Phospholipases C β Isozymes: Structural Organization and Expression in Myocardium

Among all PLC isoforms, the -β subtypes are the most extensively characterized in various cell types. The four PLCβ isozymes differ in their molecular weight: 130 kDa for PLCβ4, 140 kDa for PLCβ2, 150 kDa for PLCβ1, and 152 kDa for PLCβ3 [37]. Specifically, the -β subfamily is composed of a conserved core structure with an N-terminal PH domain, a split X-Y catalytic domain, four EF hands, a C2 domain, and an extended C-terminal domain with a PDZ domain (Figure 1A). In particular, the PH domain of PLCβ is primarily involved in regulatory protein–protein interactions, including G proteins. The EF hands serve as a structural scaffold and support for GTP hydrolysis. Moreover, unlike the C2 domains of other PLC isoforms, the C2 domains of PLCβ do not contribute to Ca^2+^-mediated interactions with the membrane. In fact, it plays a role in creating binding sites for regulatory proteins, such as Gαq regulation [38]. Additionally, only PLCβ and PLCη present the PDZ domain, which is a binding site for large molecular complexes [39].

PLCβ1, PLCβ2, PLCβ3, and PLCβ4 exhibit a relatively uniform distribution within cardiac tissue [25,27,40]. Interestingly, several studies reported that PLCβ4 has a relatively higher level of expression than PLCβ1, PLCβ2, and PLCβ3 in human, murine, and rat left ventricular (LV) tissue [25]. However, despite this observation, PLCβ1 and PLCβ3 have received more research attention on their mechanisms of activation compared to other PLCβ isozymes in the heart due to their extensive studies in other biological systems and cell types.

PLCβ activation is regulated through the classical G-protein pathway. Upon activation by specific ligands, GPCRs undergo conformational changes that allow them to activate heterotrimeric G proteins. In turn, G proteins, consisting of α, β, and γ subunits, dissociate into α and βγ subunits which play a crucial role in modulating signal transduction [9]. Historically, it has been demonstrated that PLCβ isoforms are directly activated by Gαq/11 subunits, leading to the hydrolysis of PIP_2_. Subsequently, researchers have also demonstrated that Gβγ dimer triggers inositol lipid signaling by directly activating these isozymes, except for PLCβ4 [41]. Moreover, subsequent studies have demonstrated that Rho-family small guanosine triphosphates (GTPases) are also involved in PLCβ2 activation through its recruitment to the plasma membrane (Figure 1B) [37].

## 3. Phospholipases C β Isozymes and Their Impact on the Cardiovascular System

PLCβ isozymes are a specific subgroup of PLC enzymes that have been identified to play a significant role in regulating signal transduction pathways within the cardiovascular system [36]. Indeed, several studies have provided evidence supporting that their activation triggers a cascade of events that have significant implications for cardiac functions.

It has been widely accepted that PLCβ isozymes are involved in the regulation of cardiac contractility, upon release of Ca^2+^ ions from intracellular stores [42]. The increased Ca^2+^ levels enhance the contractile force of cardiomyocytes, contributing to the strength of cardiac contractions and, consequently, maintaining the proper cardiac function and the pumping efficiency of the heart. Specifically, evidence suggested that PLCβ1 expression and activity are enhanced in diseased myocardium in both animals and humans, contributing to disease progression [43]. As a result, activation of GPCR coupled to PLCβ1 regulates DAG production, which is responsible for PKCα activation, leading to decreased contractility through sarcoplasmic reticulum Ca^2+^ depletion [43].

Importantly, the cardiac system is a highly intricate network comprising various cellular populations, including cardiomyocytes, fibroblasts, endothelial cells, smooth muscle cells, immune cells, pericytes, and stem cells [44]. Therefore, changes in PLCβ expression can impact cardiac function by exerting effects on various cardiac cell types. For instance, it has been demonstrated that higher Ca^2+^ concentrations, following PLCβ activation, play a significant role in the regulation of vascular tone, thereby inducing arterial contraction [45]. Alternatively, PLCβ enzymes also regulate endothelial cells, releasing nitric oxide (NO), which plays a crucial role in vasodilation, blood flow distribution, and the maintenance of vascular homeostasis [46]. Notably, the modulation of PLCβ isozymes has also been implicated in the intricate process of scar remodeling in myocardial infarction (MI) hearts [47]. Hence, due to the diverse effects of PLCβ isozymes on cellular heterogeneity, targeting these enzymes could hold significant promise as a potential approach not only for monitoring physiological functions but also for developing therapeutic strategies aimed at modulating cardiovascular responses.

Moreover, PLCβ isozymes have been linked to the modulation of platelet function and thrombosis [48]. Platelets play a critical role in hemostasis and clot formation, but their excessive activation can lead to unwanted thrombotic events, such as heart attacks and strokes [49]. Remarkably, researchers have provided evidence that an inherited deficiency of PLCβ2 results in bleeding and defective platelet secretion and aggregation in ex vivo models [50,51], supporting the finding that PLCβ2 is responsible for organizing the platelet cytoskeleton, which is crucial for the proper functioning of the physiological arterial system.

Based on these studies, it follows that the dysregulation of PLCβ signaling has been identified as a significant factor in various cardiovascular disorders, including cardiac hypertrophy, diabetic cardiomyopathy, and ischemia/reperfusion (I/R) injury (Figure 2). Consequently, there is a growing interest in comprehending the intricate mechanisms underlying PLCβ-related signaling pathways and exploring their potential as therapeutic targets for future interventions to address these pathological conditions.

Thus, after a brief overview of their physiologic functions, the current knowledge about the relevance of PLCβ isoforms in cardiovascular disorders is summarized below, highlighting their potential as promising diagnostic and therapeutic targets for cardiac complications.

### 3.1. Phospholipases C β Isozymes in Cardiac Hypertrophy

Cardiac hypertrophy is an adaptive response to biochemical and mechanical stimuli, such as volume or pressure overload, which are responsible for the development of congestive HF [52]. This pathological condition is characterized by an increase in cardiomyocyte size and protein synthesis resulting in LV hypertrophy. Several pieces of evidence confirmed the implication of many growth factors in triggering cardiac hypertrophy through GPCRs, which in turn mediate the activation of the PLC pathway [53]. In particular, PLCβ isozymes have been identified as pivotal players in mediating the hypertrophic response. The activation of the Gq/PLCβ pathway results in elevated cytosolic Ca^2+^ levels that trigger the activation of PKC members and calcineurin/nuclear factor of activated T cells (calcineurin/NFAT) pathway, ultimately leading to hypertrophy [53].

To elucidate the involvement of PLCβ isozymes in the hypertrophic response, researchers have explored their expression and activity by using various pro-hypertrophic stimuli, such as angiotensin II (ANGII), phenylephrine (PE), vasopressin, α1-adrenergic agonists, and other pharmacological agents. Interestingly, several lines of evidence have demonstrated the activation of PLCβ isoforms in response to these stimuli, both in in vitro and in vivo models [18,19,40,54].

Firstly, it has been observed that the PLCβ1 b splice variant is specifically upregulated in neonatal ventricular rat cardiomyocytes (NRVMs) exposed to PE stimulation, a useful drug to block β-adrenergic receptors (AR) [18,20]. Indeed, a direct correlation between PLCβ1 b expression and hypertrophic stimuli was identified, as evidenced by an increase in cell area, protein/DNA ratio, and atrial natriuretic peptide (ANP) mRNA levels. In support of this finding, the modification of the unique C-terminal tail of PLCβ1 b, which prevents Gq/PLC association, effectively abolished hypertrophic responses [18]. Additionally, these data highlight the crucial role of G proteins in modulating hypertrophy since the upregulation of Gq activity exacerbates hypertrophic induction, and, conversely, its inhibition is responsible for the enhancement. According to this study, upregulated PLCβ1 b activity was observed in diseased myocardium of mice and humans [43].

Recent studies have also highlighted the implications of all the other PLCβ isoforms in the development of hypertrophic stimuli following drug treatment. For instance, Otaegui and colleagues have shown that the administration of ANG II specifically induces a pronounced increase in PLCβ4 gene expression in HL-1 cardiomyocytes [25]. Similarly, other findings have highlighted the contribution of both PLCβ1 and -β3 in volume-overloaded rat hearts, showing a positive correlation between their expression and the early activation of atrial and right ventricular hypertrophy [23]. Indeed, PLCβ1 and -β3 silencing with siRNA resulted in a significantly attenuated norepinephrine (NE)-induced hypertrophic response due to inhibition of ANP high gene expression in LV cardiomyocytes [21]. Similarly, the use of losartan, an angiotensin II receptor blocker (ARB), has demonstrated the ability to reduce PLCβ gene expression and partially hinder the advancement of cardiac hypertrophy. This effect is substantiated by a decrease in levels of ANP as well as collagen types I and III [23]. Of note, losartan is closely related to the renin-angiotensin system (RAS), which is responsible for PLCβ activation and hypertrophic induction. Hence, these results further confirm the implication of RAS in the modulation of PLCβ expression, as displayed in hypertrophy-induced models.

In line with the abovementioned drugs, it is well known that doxorubicin (dox), a chemotherapeutic drug used for hematological and solid tumors, is responsible for irreversible cardiomyopathy characterized by cardiomyocyte damage, dilatation, and dysfunction of the left ventricle [22]. Elevated plasma levels of endothelin 1 (ET-1) and its receptors, which are associated with the hypertrophic remodeling of cardiomyocytes, have been observed in both dox-treated human [55] and animal models [56]. Interestingly, in a murine dox cardiotoxicity model, as well in mouse HL-1 cardiomyocytes, researchers have discovered a simultaneous upregulation of ET-1 and PLCβ2 [57] that was subsequently abolished by using ET-1 receptor inhibitors, thus preventing a hypertrophic response [57]. Therefore, it would be important to deepen these studies as inhibiting PLCβ2 may offer potential benefits to patients undergoing dox treatment by mitigating its adverse cardiotoxic effects.

Clearly, the modulation of PLCβ triggers a range of downstream key molecules that are crucial participants in the progression of this pathological condition. Indeed, it has been demonstrated that NE and ANG II treatments induce the upregulation of PLCβ concomitantly to a progressive increase in c-FOS and c-Jun transcription factors, suggesting a tight relation among these targets [24]. In support of this result, the use of a known PLC inhibitor, named U73122, was able to restore and attenuate their expression by inducing a cardioprotective effect [58]. Specifically, activation of PLCβ3 significantly upregulates expression levels of PKC, ERK 1/2, and Raf-1 in hypertensive rats induced by aortic restriction [40], which are controlled by PKC and ERK 1/2-dependent pathway, as demonstrated in adult cardiomyocytes by the treatment with bisindolylmaleimide and PD98059 inhibitors, respectively [58,59].

Taken together, it appears that the increase in PLCβ isozymes levels may be a common feature occurring in atrial and ventricular hypertrophic conditions [26]. Hence, these findings strongly indicate that PLCβ plays a crucial role in the induction of cardiac hypertrophy at an early stage and may additionally contribute to the persistence of the hypertrophic response, as demonstrated by pharmacological treatment.

### 3.2. Phospholipases C β Isozymes in Diabetic Cardiomyopathy

DCM is a complex heart disease that occurs as a complication of diabetes mellitus (DM). It is characterized by myocardial dysfunction in diabetic patients in the absence of apparent cardiac risk factors, such as coronary artery disease, valvular disease, and hypertension [60]. DCM has a silent progression defined by cardiomyocyte hypertrophy, cardiac fibrosis, degradation of the extracellular matrix (ECM), and mitochondrial dysfunction, which compromise cardiac stiffness, resulting in systolic dysfunction accompanied by HF with reduced ejection fraction [61]. Of note, this cardiomyopathic condition is associated with decreased contractility and intracellular Ca^2+^ transients due to a depressed cardiac SL membrane PA level, as shown in diabetic rat hearts [62].

Previous studies have provided evidence for the involvement of PLC signaling pathways in the development of insulin-dependent cardiomyopathy; indeed, this cardiomyopathy is associated with a disruption in α1-AR stimulation [63], which, in turn, affects the activity of PLCβ isozymes via Gqα [64]. Specifically, the diabetic state may impact PLCβ3 signaling within cardiomyocytes. Primarily, it affects PLCβ3 distribution within cardiomyocytes, and, further, it modifies the coupling between Gqc and PLCβ3 in response to α1-adrenergic stimulation, as demonstrated in insulin-treated rat hearts [65]. Hence, a decrease in IP_3_ levels is correlated with a reduced PLCβ3 activity, which may contribute to a decrease in the strength of contraction of the isolated papillary muscle in response to α1-adrenergic stimulation [63]. Furthermore, a reduction in the production of PLCβ3-derived DAG can have significant effects on various cellular processes since it has been demonstrated to affect downstream targets, such as PKC isoforms and other regulators, involved in Ca^2+^ transient fluxes [28,66]. For instance, Ruboxistaurin, a PKC-β inhibitor, has demonstrated its efficacy in both animal and human clinical trials for treating diabetic vascular complications. Indeed, it effectively preserves cardiac function and reduces structural injury, highlighting its promising potential as a therapeutic intervention [67].

Additionally, diabetic cardiomyopathy is characterized by an increase in oxidative stress levels, and, considering the inhibitory effect of oxidants on PLC [68], it is possible that the reduction in PLCβ3 levels observed in diabetic cardiomyopathic conditions may be attributed to the induction of oxidative conditions. Consequently, employing antioxidants may emerge as a promising strategy to effectively counteract and mitigate the effects of diabetic cardiomyopathy.

Thus, the involvement of the PLC signaling pathway shown by the reduced PLCβ3 activity, along with increased oxidative stress, can be considered one of the key factors in the development and progression of DCM by influencing downstream signaling events and processes involved in cardiac contractility [65]. Definitely, additional investigations will provide deeper insights into the specific molecular pathways, potential therapeutic targets, and strategies to mitigate the adverse effects of altered PLCβ3 signaling in diabetic individuals.

### 3.3. Phospholipases C β in Myocardial Ischemia/Reperfusion Injury

I/R injury is a leading cause of myocardial damage characterized by a reduced blood supply to organs followed by the subsequent restoration of perfusion and oxygen supply [69]. Despite the reestablished coronary flow being beneficial for the restoration of cardiac activity, it has been demonstrated that the reperfusion, following a specific period of ischemia, can further exacerbate myocardial anomalies [70]. Of note, I/R injury is characterized by a cellular energy depletion that disrupts the activity of ATP-dependent calcium pumps, such as the sarcoplasmic reticulum calcium ATPase (SERCA), responsible for calcium reuptake into the sarcoplasmic reticulum [71]. As a consequence, higher intracellular Ca^2+^ levels trigger the activation of calcium-dependent enzymes, such as PLC and proteases, promoting various detrimental effects. Among them, I/R injury results in the induction of oxidative stress, energy metabolism disorder, cardiomyocyte death, and autoimmune response, potentially leading to long-term cardiac dysfunction [70,71,72]. Therefore, the development of effective strategies to minimize I/R injury is crucial to decrease and prevent adverse effects in patients.

As mentioned above, I/R injury triggers several protein kinase families, such as PKC isoforms, phosphoinositide 3-kinase (PI3K)/protein kinase B (AKT)/mammalian target of rapamycin (mTOR) axis, and mitogen-activated protein kinase (MAP) kinases, which are the downstream target of PLC isozymes [73]. However, contrasting findings have been reported on the impact of I/R on PLC activity, highlighting the complex and multifaceted nature of PLC regulation and the necessity for further and in-depth investigations. Indeed, some studies have shown variations in PLC metabolism under ischemic conditions, with both increased and decreased activity [74]. On the other hand, during the reperfusion phase, there is a notable increase in PLC activity [75,76].

Interestingly, researchers have highlighted the potential implication of PLCβ1 as a marker of I/R conditions due to a significant increase in PLCβ1 activity in the ischemic heart, which undergoes a progressive depletion during the reperfusion phase [77]. Supporting this evidence and recognizing the role of PLCβ in Ca^2+^ overload, compelling studies have demonstrated the effectiveness of α1 receptor antagonist prazosin [29] and L-type Ca^2+^-channel blocker verapamil [78] in reducing I/R injury through the inhibition of PLCβ1. This effect is achieved by partially inhibiting the increase in PLCβ1 activity during ischemia and preventing its decline during the reperfusion phase. Notably, the specific activation of PLCβ1 plays a crucial role in triggering the activation and translocation of the downstream PKCε isozyme, which mediates the ischemic preconditioning, a cardioprotective adaptation against prolonged ischemic events [30]. More specifically, PKCε binds to LCK, a member of the Src family of tyrosine kinases, and prevents myocardial infarction [79] by activating MAPK pathways [80,81], which in turn limits the accumulation of cytosolic Ca^2+^ during I/R injury.

In summary, few studies have been performed on this subfamily of enzymes in I/R injured patients, but promising evidence suggests that inhibiting PLCβ1 activity can be a potential strategy to reduce I/R injury and activate cardioprotective pathways.

## 4. Conclusions

This review emphasizes the compelling evidence supporting the potential of PLCβ isozymes as biomarkers in the context of cardiovascular disease. It suggests a strong correlation between imbalances in PLCβ activity and expression and the development of various cardiac conditions, which are still mostly unclear and require further in-depth investigations. Currently, studies in the literature have revealed significant activation of this subfamily in cardiac hypertrophy, diabetic cardiomyopathy, and ischemia/reperfusion injury. However, a complete understanding of the intricate network of PLCβ with its downstream targets and signaling pathways demands further research in both healthy and diseased myocardium. Based on the studies carried out so far, these isoenzymes hold promise as potential diagnostic markers and therapeutic targets for the previously mentioned cardiac pathologies. Including PLCs as additional biomarkers alongside established ones has the potential to significantly enhance diagnostic accuracy and provide a more comprehensive clinical perspective due to their involvement in critical cellular mechanisms. Inhibiting the PLCβ subfamily using different approaches, such as pharmacological inhibitors or genetic interventions, can represent a promising strategy to manipulate the outcome of cardiovascular disorders. Moreover, the translation of insights from in vitro and animal models into clinical applications is crucial for understanding PLCβ isoforms’ relevance. Indeed, the conceivable physiological and metabolic differences between animal models and human patients underline the need for comprehensive clinical studies that should encompass diverse patient cohorts, considering severity and disease variability, in order to increase the current knowledge. Hence, filling this gap is essential to thoroughly establish the clinical significance of PLCβ isoforms and to exploit their potential in future diagnosis and treatment of cardiovascular pathologies. Therefore, further research on PLCβ signaling pathways is necessary to pave the way for potential therapeutic strategies, aimed at effectively attenuating cardiac side effects in patients.

## Figures and Tables

**Figure 1 ijms-24-13096-f001:**
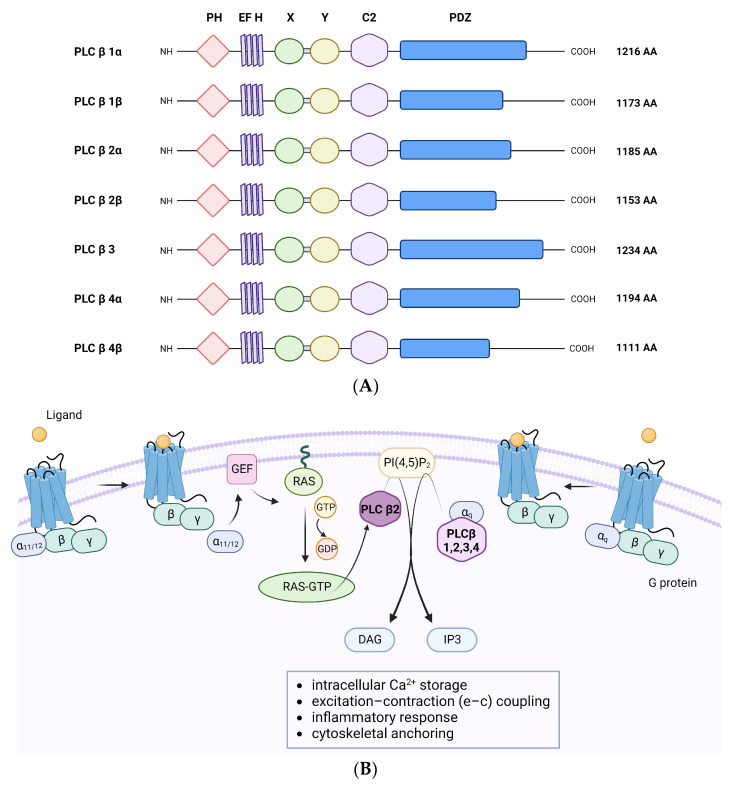
Domain organization of PLCβ isozymes and their activation mechanisms. (**A**) Each isoform has a catalytic core composed of an N-terminal PH domain, four EF-hand motifs, an X–Y catalytic domain, a C2 domain, and a C-terminal domain with a PDZ binding motif. The lengths and sequences of the C-terminal domains can diverge among different isoforms and splice variants. (**B**) PLCβ isozymes are activated by GPCRs, leading to the production of important second messengers, such as IP_3_ and DAG. Additionally, PLCβ2 can also be activated by small GTPases of the Rho family, such as Rac, which is responsible for its recruitment at the plasmalemmal membrane. Phospholipase C β (PLCβ); G-protein-coupled receptors (GPCRs); phosphatidylinositol-4,5-biphosphates (PIP_2_); hydrophilic acidic end-group inositol (1,4,5) trisphosphate (IP_3_); neutral lipid sn-1,2-diacylglycerol (DAG); guanine nucleotide exchange factor (GEF); guanosine-5’-triphosphate (GTP); guanosine diphosphate (GDP).

**Figure 2 ijms-24-13096-f002:**
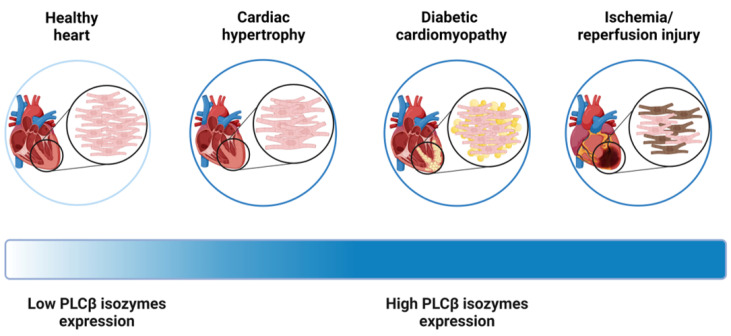
Expression of PLCβ in normal and diseased myocardium. The basal expression of PLCβ in the myocardium is characterized by low levels, but it undergoes significant alterations during pathological cardiac conditions, including cardiac hypertrophy, diabetic cardiomyopathy, and ischemia-reperfusion injury. These clinical scenarios are associated with increased PLCβ expression.

**Table 1 ijms-24-13096-t001:** Summary table of PLCβ isoforms implicated in cardiac pathologic conditions.

Pathologic Condition	PLCβ Isoform	Model
Hypertrophy	PLCβ1	Rat neonatal cardiomyocytes [18,19], Sprague-Dawley rats [20,21,22], neonatal rat ventricular myocytes [23]
	PLCβ2	C57BL/6N mice [24], HL-1 murine cardiomyocytes [24]
	PLCβ3	Sprague-Dawley rats [25], neonatal rat cardiomyocytes [26]
	PLCβ4	Human left ventricle biopsy [27], Wistar-Kyoto rats [27], BALB/c mice [27], HL-1 murine cardiomyocytes [27]
Diabetic cardiomyopathy	PLCβ3	Sprague–Dawley rats [28]
Ischemia/reperfusion injury	PLCβ1	Sprague–Dawley rats [29,30]

## Data Availability

No new data were created or analyzed in this study.

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
