# Peer review of "Emerging Roles of Phospholipase C Beta Isozymes as Potential Biomarkers in Cardiac Disorders"

_ijms, 2023, doi:10.3390/ijms241713096_

Round 1

Reviewer 1 Report

In this review, the authors work up the role of Phospholipase C Beta in different cardiac disorders like diabetic cardiomyopathy or ischemia/reperfusion injury or cardiac hypertrophy. This review suggests a strong activation of the Isozyme PLCbeta in those cardiac disorders, although the intricate network of PLCbeta with its downstream targets and signalling pathways are not fully understand yet. All in all, a lot of information are collected in this review (also a lot of textbook wisdom), which might be interesting in the context of understanding the pathomechanism of those disease, but I am not sure about their clinical relevance.

In general, the topic is of high relevance for the readership of The International Journal of Molecular Sciences and I really appreciate the informative figures in this review. There are some small comments and questions for the authors which are listed below. Nevertheless, I would recommend to accept this manuscript with minor revisions.

-        Is there a connection between the clinical outcome of patients and the PLCbeta isozyme, indicating it as a good biomarker? Is there any potential therapeutic tool described which influencing the PLCbeta isozymes?...I am just thinking about the use in a clinical setting… which benefit might result from measuring PLCbeta in patients?

-        I would recommend to shorten the introduction and the first section about the myocardial phospholipases C isozymes, because this review only focus on the Beta isoform

-        Is there any information about acute cardiac diseases and the role of Phospholipase C Beta for example in myocardial infarction or cardiac contusion?

-        Small corrections:

o    L 43: “…cytoskeletal anchoring.[1].”

o    L63 “…signaling cascade” The literature is missing

o   Figure 1: Please explain the abbreviation, when using them the first time (also in the figure legend)

Author Response

In this review, the authors work up the role of Phospholipase C Beta in different cardiac disorders like diabetic cardiomyopathy or ischemia/reperfusion injury or cardiac hypertrophy. This review suggests a strong activation of the Isozyme PLCbeta in those cardiac disorders, although the intricate network of PLCbeta with its downstream targets and signalling pathways are not fully understand yet. All in all, a lot of information are collected in this review (also a lot of textbook wisdom), which might be interesting in the context of understanding the pathomechanism of those disease, but I am not sure about their clinical relevance.

In general, the topic is of high relevance for the readership of The International Journal of Molecular Sciences and I really appreciate the informative figures in this review. There are some small comments and questions for the authors which are listed below. Nevertheless, I would recommend to accept this manuscript with minor revisions.

-        Is there a connection between the clinical outcome of patients and the PLCbeta isozyme, indicating it as a good biomarker? Is there any potential therapeutic tool described which influencing the PLCbeta isozymes?...I am just thinking about the use in a clinical setting… which benefit might result from measuring PLCbeta in patients?

We really thank reviewer 1 for the critical analysis, which significantly enhances the relevance of our research. However, in recent years, accumulating evidence has increasingly linked cardiovascular disorders with the activity and expression of PLCβ isozymes, suggesting their potential role in early disease development and ongoing pathological processes. For instance, a higher expression of these enzymes corresponds to a more severe hypertrophic condition; however, only a limited number of patient investigations have been conducted. Certainly, to establish PLCβ isozymes as reliable biomarkers, future clinical studies could focus on assessing the expression levels and activity of specific PLCβ isoforms in patient cohorts with varying degrees of cardiovascular diseases. Studies tracking changes in PLCβ isozyme profiles alongside clinical outcomes would provide valuable insights into the biomarker potential of these enzymes. Currently, some studies have explored small molecule inhibitors or modulators that can influence the activity of specific PLCβ isoforms, demonstrating the attenuation of aberrant signaling pathways associated with cardiac diseases. Therefore, manipulating the levels of these enzymes might help restore cellular signaling balance and mitigate the pathological processes observed in cardiovascular diseases. Hence, measuring PLCβ isoforms in patients could potentially offer several benefits in a clinical setting, including early disease detection, as alterations in PLCβ isoforms may occur before overt clinical symptoms manifest, and treatment monitoring, as changes in PLCβ profiles might serve as indicators of treatment response or disease progression. However, it's important to emphasize that the development of therapeutic tools targeting PLCβ isozymes is still in its early stages, and further research is needed to evaluate their safety and efficacy in preclinical and clinical settings.

-        I would recommend to shorten the introduction and the first section about the myocardial phospholipases C isozymes, because this review only focus on the Beta isoform

We followed the suggestion and shortened the length of the sections.

-        Is there any information about acute cardiac diseases and the role of Phospholipase C Beta for example in myocardial infarction or cardiac contusion?

According to a paper published in 1998, there is only one evidence demonstrating the upregulation of PLCβ1 and PLCβ3 expression in the border and scar tissues of post-myocardial infarction rat hearts (https://doi.org/10.1161/01.CIR.97.9.892); afterward, no other studies have been conducted in this specific cardiac field. Regarding cardiac contusion, there are no studies in literature.

-        Small corrections:

o    L 43: “…cytoskeletal anchoring.[1].”

o    L63 “…signaling cascade” The literature is missing

o   Figure 1: Please explain the abbreviation, when using them the first time (also in the figure legend)

As kindly suggested by Reviewer 1, we corrected all the information that were inaccurate or uncomplete.

Reviewer 2 Report

In this review, Fazio and colleagues delve into the potential of Phospholipase C (PLC) beta isozymes as indicators in cardiac disorders. The authors underscore the profound influence of PLC on the cardiovascular system, both under normal physiological conditions and in disease states. They mainly focus on PLCβ variants, highlighting their role in various cardiac pathologies such as cardiac hypertrophy, diabetic cardiomyopathy, and myocardial ischemia/reperfusion injury. The authors propose that a deeper understanding of the regulatory mechanisms of PLCβ could pave the way for developing targeted therapeutic strategies for cardiac conditions. However, the article could be improved in the following ways:

  1. A more detailed exploration of the limitations of current research and the challenges faced in translating these findings into clinical practice would provide a more balanced perspective to the readers.
  2. While the authors extensively discuss the potential of PLC beta isozymes as biomarkers, providing practical examples or case studies where these biomarkers have been successfully applied in a clinical setting would enhance the article's relevance.
  3. How do PLCs compare to other potential biomarkers?
  4. The authors have based their review on studies conducted on diverse models (in vitro, in vivo, human, and animal models). It's crucial to acknowledge that findings from animal or in vitro studies may not directly apply to humans due to physiological and metabolic differences.

Author Response

In this review, Fazio and colleagues delve into the potential of Phospholipase C (PLC) beta isozymes as indicators in cardiac disorders. The authors underscore the profound influence of PLC on the cardiovascular system, both under normal physiological conditions and in disease states. They mainly focus on PLCβ variants, highlighting their role in various cardiac pathologies such as cardiac hypertrophy, diabetic cardiomyopathy, and myocardial ischemia/reperfusion injury. The authors propose that a deeper understanding of the regulatory mechanisms of PLCβ could pave the way for developing targeted therapeutic strategies for cardiac conditions. However, the article could be improved in the following ways:

  1. A more detailed exploration of the limitations of current research and the challenges faced in translating these findings into clinical practice would provide a more balanced perspective to the readers.

We really thank reviewer 2 for the critical analysis and the suggestions, which significantly enhance the relevance of our research. As suggested, we added some critical comments in the conclusions of the manuscript (Lines: 375-387).

2. While the authors extensively discuss the potential of PLC beta isozymes as biomarkers, providing practical examples or case studies where these biomarkers have been successfully applied in a clinical setting would enhance the article's relevance.

Currently, investigations have been limited to the assessment of PLC beta isozyme expression in affected human heart tissues. Indeed, the primary focus of the review is to offer a comprehensive overview of the research landscape surrounding PLCβ isozymes in this field. Although clinical instances or case studies illustrating their clinical application are not currently accessible, the review is designed to synthesize the existing evidence, highlight potential areas of clinical relevance, and identify directions for future research.

3. How do PLCs compare to other potential biomarkers?

Including PLCs as additional biomarkers alongside established ones, such as ANP, BNP, troponin and others, may hold significant potential to enhance diagnostic accuracy and provide a more comprehensive clinical perspective. Indeed, their distinct advantage lies in their involvement in critical cellular signaling pathways, which are crucial to cardiovascular health, as highlighted in the review.

4. The authors have based their review on studies conducted on diverse models (in vitro, in vivo, human, and animal models). It's crucial to acknowledge that findings from animal or in vitro studies may not directly apply to humans due to physiological and metabolic differences.

Taking in consideration the suggestions, we discussed this evidence in the conclusions of the manuscript (Lines: 375-387).

Reviewer 3 Report

I enjoyed reading this well-structured review article by Fazio et al. The authors provide a comprehensive overview of the role of phospholipase C (PLC) enzymes and their isoforms implications in different cardiac pathological conditions. The potential of PLCβ isoforms as cardiac biomarkers and therapeutic targets is also well discussed.

General comments (in no order of magnitude)

·        The authors briefly discussed the role of PLCβ in non-myocyte cells of the heart, (line-177) such as fibroblasts, and endothelial cells. They should also include this article by Ju et al, where they explain PLCβ during fibrotic remodelling (PMID: 9521338).

·        Can authors consider making an abstract figure or summary table showing different cardiovascular disorders and their associated PLCβ isoform?

            Differential regulation of phospholipase C-beta isozymes in cardiomyocyte hypertrophy (PMID: 10944430)

            Phospholipase C gene expression, protein content, and activities in cardiac hypertrophy and heart failure due to volume overload (PMID: 15072958)

            Phospholipase C signaling tonically represses basal atrial natriuretic factor secretion from the atria of the heart (PMID: 23479262

·        Check if the reference 14 is correct.

Author Response

I enjoyed reading this well-structured review article by Fazio et al. The authors provide a comprehensive overview of the role of phospholipase C (PLC) enzymes and their isoforms implications in different cardiac pathological conditions. The potential of PLCβ isoforms as cardiac biomarkers and therapeutic targets is also well discussed.

General comments (in no order of magnitude)

  • The authors briefly discussed the role of PLCβ in non-myocyte cells of the heart, (line-177) such as fibroblasts, and endothelial cells. They should also include this article by Ju et al, where they explain PLCβ during fibrotic remodelling (PMID: 9521338).

We really thank reviewer 3 for the great effort and attention put in the revision of our manuscript. We included the article suggested by the reviewer. 

  • Can authors consider making an abstract figure or summary table showing different cardiovascular disorders and their associated PLCβ isoform?

As suggested by reviewer 3, the detailed summary table of PLCβ isoforms correlated to cardiovascular disorders was included in the introduction.

  • Differential regulation of phospholipase C-beta isozymes in cardiomyocyte hypertrophy (PMID: 10944430)
  • Phospholipase C gene expression, protein content, and activities in cardiac hypertrophy and heart failure due to volume overload (PMID: 15072958)
  • Phospholipase C signaling tonically represses basal atrial natriuretic factor secretion from the atria of the heart (PMID: 23479262

As kindly suggested by reviewer 3, we included all the suggested articles.

  • Check if the reference 14 is correct.

We apologize for the mistake, and we have replaced it with the correct reference.

Round 2

Reviewer 2 Report

The authors have addressed all concerns satisfactorily.